# Flake Graphene as an Efficient Agent Governing Cellular Fate and Antimicrobial Properties of Fibrous Tissue Engineering Scaffolds—A Review

**DOI:** 10.3390/ma15155306

**Published:** 2022-08-02

**Authors:** Aleksandra Izabela Banasiak, Adrian Racki, Marcin Małek, Adrian Chlanda

**Affiliations:** 1Graphene and Composites Research Group, Łukasiewicz Research Network—Institute of Microelectronics and Photonics, Al. Lotników 32/46, 02-668 Warsaw, Poland; adrian.racki@imif.lukasiewicz.gov.pl (A.R.); adrian.chlanda@imif.lukasiewicz.gov.pl (A.C.); 2Research Laboratory of WIG, Faculty of Civil Engineering and Geodesy, Military University of Technology, 2 Gen. S. Kaliskiego St., 00-908 Warsaw, Poland; marcin.malek@wat.edu.pl

**Keywords:** electrospun scaffold, polymeric biomaterials, antimicrobial properties, graphene modifications, tissue engineering

## Abstract

Although there are several methods for fabricating nanofibrous scaffolds for biomedical applications, electrospinning is probably the most versatile and feasible process. Electrospinning enables the preparation of reproducible, homogeneous fibers from many types of polymers. In addition, implementation of this technique gives the possibility to fabricated polymer-based composite mats embroidered with manifold materials, such as graphene. Flake graphene and its derivatives represent an extremely promising material for imparting new, biomedically relevant properties, functions, and applications. Graphene oxide (GO) and reduced graphene oxide (rGO), among many extraordinary properties, confer antimicrobial properties of the resulting material. Moreover, graphene oxide and reduced graphene oxide promote the desired cellular response. Tissue engineering and regenerative medicine enable advanced treatments to regenerate damaged tissues and organs. This review provides a reliable summary of the recent scientific literature on the fabrication of nanofibers and their further modification with GO/rGO flakes for biomedical applications.

## 1. Introduction

The significant development of nanotechnology in the recent decades allows the design and preparation of fibrous nanomaterials with applications in many fields [1,2]—nanofibers are extremely promising materials, especially taking into account their unique physical, chemical, mechanical, and biological properties, which can be advantageous regarding medicine, bioengineering, and tissue engineering [3]. Nanostructured materials are willingly used as carriers for guided drug delivery [3,4,5], in tissue engineering as cellular scaffolds [6,7,8,9], in regenerative medicine as dressings to accelerate regeneration of damaged skin or tissues [10,11,12]. It is worth underlining that reducing the diameter of fibers to nanometer size significantly increases the surface area to volume ratio of the material, thus increasing its ability to retain fluids [13,14].

Several brilliant reviews concerning tissue engineering scaffolds have been published [15,16], but there is a lack of comprehensive summary of graphene influence on antimicrobial properties of fibrous tissue engineering scaffolds. This work is a reliable review of the newest scientific papers focusing on nanofibrous materials prepared by electrospinning for bioapplications in medicine and tissue engineering. As such, we decided to present scientific reports from recent years that address above all the improved biological and antimicrobial properties of electrospun fibrous materials, which were functionalized by the addition of graphene oxide (GO) or reduced graphene oxide (rGO). Graphene, a comparatively novel 2D carbonaceous material, presents very interesting and unique properties which might be appealing for previously mentioned fibrous scaffolds, leading to lower the chance for infection and higher cellular adhesion [17]. We summarized the state of research on antimicrobial and antiviral activities of GO/rGO, highlighting any inaccuracies and underlining questions for further studies. The subject of this manuscript gives the reader an opportunity to follow not only recently developed composite materials and their properties, but also utilization of such materials as an efficient tool of personalized modern medicine [17,18,19]. Based on previous studies, we showed that composite fibrous mats, enhanced with flake graphene are characterized with antimicrobial properties and superior cellular behavior, including promotion of human cell growth and proliferation. Lastly, we described challenges of graphene-based materials utilization and emphasized the importance of developing new scientific methods in order to effectively characterize such a 2D material and evaluate its influence on selected properties of fibrous scaffolds in tissue engineering, especially regarding their surface.

## 2. Electrospinning as a Satisfactory Method for Producing Micro- and Nano-Sized Fibers

Electrospinning is a feasible, low-cost and versatile method that allows the fabrication of nanofibers using different types of materials [20,21,22,23]. Moreover, this method can be affordably modified to obtain core/shell structured fibers [20]. Electrospun fibrous scaffolds are characterized by among others: fiber homogeneity (diameter of single fiber), porosity, and high surface area to volume ratio [20,24]. The electrodeposition process can be modified by varying the current, operating the working distance, or collector rotation speed. At the same time, it should be mentioned that the main drawbacks of electrospinning, precluding from its large-scale deployment are related with the long time needed to prepare a sufficient amount of fibrous material and the small production rate [25].

Electrospinning is a well-known method of controllable fabrication of nanofibers that is driven by an external voltage [26,27]. The main components of the electrospinning setup are an injection pump, a metal needle—which acts as an electrode, a high voltage power supply, and a grounded collector. Figure 1 shows a schematic illustration of the apparatus for fabrication of nanofibrous materials by electrospinning.

Individual polymeric fibers are formed and stretched under the influence of the electrostatic field acting between the needle and the collector. The solvent evaporates and the spliced fibers are expanded and then accumulated on the collector [29]. The same rule applies to composite fibers.

## 3. Graphene—Properties, Synthesis, and Applications

Graphene (G), allotrope of sp2 hybridized carbon atoms with unusual electric [30], thermal [19], and mechanical [17] properties has attracted enormous interest in the past decade [31]. Methods of its preparation can be divided into two categories: top-down (graphene is extracted from other carbon source through sonication, chemical or mechanical exfoliation) and bottom-up (building a graphene structure atom by atom using techniques such as chemical vapor deposition CVD) [32]. The first of them leads to large quantities of graphene derivatives: graphene oxide GO or reduced graphene oxide rGO [33]. The bottom-up method (such as CVD) is used to grow monolayers of graphene on specific transition metal substrates [34]. This two-dimensional material has many possible applications. Some of them (such as masks [35] or concrete with graphene [36]) are already commercially available. Others, for instance, supercapacitor devices based on graphene materials [37] or graphene oxide as a scaffold for bone regeneration [38,39,40], are still in the research phase, waiting for their fruitful technical validation. However, large-scale production of cheap and high-quality graphene is a bottleneck of the whole development and research process. A solution to the profitability issue and advantage for graphene utilization can be the synthesis of GO/rGO from waste materials, such as waste toner powder [41], waste newspaper [42], and oil palm waste (kernel shells, leaves, and empty fruit bunches) [43].

The most popular derivatives of flake graphene are GO and rGO (Figure 2). GO has a structure similar to the theoretical structure of graphene, although the carbon layer is densely decorated with oxygen functional groups, such as hydroxyl (-OH), carboxyl (-COOH) or alkoxy (C-O-C) groups [32]. Having in mind well-established chemical synthesis of GO, this is the direct outcome of aggressive reaction conditions [44,45] of graphite oxidation and exfoliation. Due to presence of those oxygen functional groups, GO has different properties compared to rGO or G [46]. Pristine graphene is a highly thermally $\left(1800\;{W\;m}^{-1}K^{-1}-5000{\;W\;m}^{-1}K^{-1}\right)$ and electrically conducting material with high electron mobility $\left(2.5\;m^{2}V^{-1}{\;s}^{-1}\right)$ [47,48,49] due to the zero-overlap semimetal with electron and holes as charge carriers [50]. It was reported in the literature that graphene addition to other materials can improve their thermal and electrical conductivity [51,52]. Preparation of GO flakes results in addition of functional groups to the carbon lattice causing disruption of the sp2 bonding orbitals of graphene. This is the key mechanism governing GO’s thermal $\left(0.14-18{\;W\;m}^{-1}K^{-1}\right)$ and electrical properties $\left(2.5\times 10^{3}-25\times 10^{3}\;\Omega m\right)$ [53,54,55]. Moreover, GO can be easily dispersed in water and some organic solvents [56]. GO presents high surface functionalization potential with other functional groups such as nitrogen, sulfur, and phosphorous [57]. For example, recent research suggests that sulfonated graphene oxide (graphene oxide with substituted sulfonic acid functional groups) presents possible use as a catalyst in esterification and transesterification reactions [58,59]. A number of authors have also recognized its potential applications in medicine as drug delivery system [60,61]. Studies have shown that mechanical properties of GO are significantly worse than pristine graphene. However, they are still good enough to consider GO flakes as a composite nanofiller to bolster their mechanical properties. A study by Suk et al. [62] concluded that a monolayer of graphene oxide had an effective Young’s modulus of 207.6 ±23.4 GPa (compared to the theoretical 1 TPa for graphene [17]). For example, researchers have provided evidence that GO addition to asphalt [63] and cement [64] can improve their mechanical performance and chemical resistance.

The reduction process leading to synthesis of rGO from GO causes removal of most of the oxygen functional groups and can create some defects in the atom-thick structure of carbon atoms [46]. The reduction reaction itself can be achieved by treating GO with chemicals (e.g., hydrazine and its derivatives) [65], microwaves [66] or high temperatures [67]. Thanks to that, rGO has very similar structure and properties to pristine graphene and has relatively low cost of overall synthesis [68]. Contrary to GO, rGO is highly conductive $\left(2\times 10^{2}{\;S\;m}^{-1}\right)$ [69] and hydrophobic [70]. Nonetheless, rGO still possesses oxygen-containing functional groups which can be used to further functionalization. In addition, the amount of remaining functional groups can be controlled via different reduction techniques. Thanks to resemblance to graphene, it was reported that rGO flakes have many potential applications. A large number of existing studies have depicted rGO introduction in the detection of specific substances, molecules or atoms at low concentrations [71,72,73]. A number of authors have also recognized that rGO can improve the stability and safety of materials used to store electric energy [74].

As this review is focused on the biomedical application of flake graphene, subsequent paragraphs will be devoted exclusively to flake graphene properties closely related with this subject.

## 4. Antimicrobial and Antiviral Activity of Graphene Oxide GO and Reduced Graphene Oxide rGO

### 4.1. Toxicity towards Bacteria and Fungi

Flake graphene is a promising and widely studied material in the field of bioengineering and novel biomaterials (Figure 3). There have been numerous studies aimed to investigate graphene influence on living organisms. This section presents a review of the literature on antimicrobial activity of graphene derivatives and mechanisms in which this activity occurs. Most of the papers have been focused on GO or rGO flakes and their interaction with bacteria such as Gram-negative *Escherichia coli* (*E. coli*) [75,76,77,78,79,80,81]. As it can be seen in Table 1, all of the revised studies confirmed antimicrobial activity of GO and/or rGO. The loss of viability was higher when GO was used, compared to rGO. Generally, longer exposure to the material and its higher concentrations improved antimicrobial activity. Apart from studies performed using GO and rGO, Liu et al. [75] took into an investigation two other carbon-based materials, namely graphite and graphite oxide (GO before ultrasound exfoliation) and confronted it against *E. coli.* Most of the cells lose their cellular integrity after contact with GO/rGO (Figure 4). In an isotonic saline solution, the GO suspension showed the highest death rate of bacteria, followed by rGO, graphite and graphite oxide. As expected, the study proved as well that longer incubation time and higher concentration of GO/rGO improve the antibacterial properties towards *E. coli* bacteria. It should be underlined that the lateral size of GO sheets plays a significant role in inhibition on the viability of bacteria cells [77]. Smaller flakes of graphene oxide caused higher a death rate of the tested bacteria. Sizes from 0.1 ${\mu m}^{2}$ to 0.65 ${\mu m}^{2}$ were controlled by time of ultrasound sonication of GO nanosheets. Nevertheless, one study reported the complete opposite [82]. Thus, further studies and systematically screening are needed to fully understand this phenomenon. Other species of bacteria were studied in the same manner to determine whether graphene antibacterial activity is cell-dependent. For example, in one study, the loss of *Pseudomonas aeruginosa* viability was almost the same comparing the GO and rGO suspension [78]. In the contrary, GO caused a much higher death rate among *Ralstonia solanacearum* [79] and *Xanthomonas oryzae pv. Oryzae* [80] cell culture than rGO. Chen et al. tested graphene suspensions in different buffers, i.e., DI water, 0.9% NaCl and 0.1 M phosphate-buffered saline PBS. Whereas DI water and NaCl suspensions did not affect the viability loss, PBS completely suppressed antibacterial activity of GO and rGO, and it was explained by the possible aggregation of flakes in the salt-rich medium. Paper made from GO or rGO exhibited antibacterial properties towards *E. coli*, which showed that water dispersion is not needed [76].

Antifungal effects of rGO were studied using *A. niger*, *A. oryzae*, *F. oxysporu.* After a 7-day incubation, growth of all fungi were completely stopped in solutions where rGO concentration was higher than 250 μg/mL [81] (Figure 5). In their study, Al-Thani and co-authors [83] prepared GO by a modified Hummers method and characterized it by different techniques. The XRD analysis of graphite powder showed a highly ordered structure, which corresponds to an interlayer spacing of about 0.335 nm. To study the antifungal activities of GO, the material was tested against eukaryotic fungus—*Candida albicans*. This eukaryotic fungus is characterized by a cell structure and metabolism hard to suppress by any antimicrobial agent. In this study, the loss of viability increased with incubation time of the analyzed microorganisms. Results revealed that GO has antifungal activity against microorganisms used in this investigation. To summarize, the developed GO exhibited excellent antifungal properties. Sawangphurk and co-workers [81] emphasized that antibacterial activities of graphene and its derivatives had been sufficiently investigated but their antifungal properties were far less studied. In their work, they studied the antifungal activity of rGO against three fungi: *Aspergillus oryzae*, *Aspergillus niger*, *Fusarium oxysporum*. The authors highlighted that graphene (and its derivatives) is of interest due to its high surface area (about 2630 m^2^ g^−1^), high electrical conductivity (about 2000 S cm^−1^), high thermal conductivity (about 4840–5300 W m^−1^ K^−1^), and high Young’s modulus (about 10 TPa) what leads to various potential applications. Reduced graphene oxide (rGO) was produced with a modified Hummers method. The graphene was dispersed in agar and poured into sterilized Petri dishes. Agar discs were covered with fungal; next, discs were placed aseptically in the center of agar plates containing rGO nanosheets (0–500 μg mL^−1^). Experiments were performed for 7 days in five replicates. The average diameters of fungal colonies were determined. The growth inhibition of *A. niger*, *F. oxysporum* and *A. oryzae* was proportional to the concentrations of rGO flakes. The reason that the rGO flakes were effective to inhibit fungi was probably due to the direct contact with the cell walls of the analyzed fungi. Scientists reported [81] that rGO inhibited the mycelial growth of the fungi and they hypothesized that it was due to its sharp edges. Upon contact, the reactive oxygen-containing functional groups in several small rGO nanospheres were able to chemically react with the organic functional groups of chitin and other polysaccharides on the fungal cell walls. The half maximal inhibitory concentration (IC50) is a measure of the effectiveness of the rGO in inhibiting the fungi. Authors reported that IC50 values of the rGO against *F. oxysporum*, *A. niger*, and *A. oryzae* were 50, 100, and 100 μg mL^−1^, respectively. According to the results, the fungitoxicity of rGO against the analyzed pathogenic fungi might support the possibility of using rGO as an antifungal nanomaterial.

### 4.2. Cytotoxicity of Graphene Derivatives

Cytotoxicity of graphene materials was also studied and GO induced lower loss of mammalian cell viability than rGO [76]. It is worth noting that lower cell viability occurred due to decreased proliferation rates, not to apoptosis or death of cells like in the case of bacteria. Toxicity of graphene materials in in vitro and in vivo studies was broadly reviewed in Lalwani et al. [84]. It was reported that it is highly dependent on parameters such as time, cell type, size, purity, amount of oxygen functional groups, and morphology of graphene. Even though the majority of studies show that GO and rGO flakes are cytotoxic towards bacteria and fungi, the overriding conclusion might be that specific applications will need separate studies and previous research might be used only as a guide. In fact, it is easy to notice that opposite opinions on graphene cytotoxicity have emerged. Further studies are definitely needed in order to set new frames of knowledge regarding the aforementioned subject.

The authors of [85] highlighted that despite of many advantages and exciting results of using graphene in biomedical engineering and tissue engineering, there may still be a long way ahead before the actual application of this material in clinical practice. The further biological applications of graphene have been often challenged by concerns regarding its potential cytotoxicity. However, the authors of [85] emphasized that graphene, with “nano-small” sizes, subjected to adequate purification methods, can be implemented as a biocompatible surface coating and is characterized with brilliant stability in physiological environments, seems to be much less damaging regarding both in vitro and in vivo studies. Moreover, the future focus should be placed on research leading to answers on how to abolish toxicity and affect the degradation of graphene in biological systems of living organisms. Before graphene and graphene-enriched materials can be used in clinical practice, complex studies are needed to resolve such safety issues.

### 4.3. The Antimicrobial Mechanisms of GO and rGO

Many different mechanisms describing the antimicrobial effects of GO and rGO have been proposed. Among them, the most important are membrane stress and oxidative stress [86]. Physical damage to the cell body leading to the content leakage and death of the cell is the most obvious one. SEM and TEM images of dead cells seeded on GO/rGO show irreversible damage to the cell membranes and suggest that they have been “cut” by thin layers of graphene. This theory has been supported by size- and dispersion-dependency of the material. As mentioned before, GO is hydrophilic and thus easily dispersible in water [56]. Studies suggest that GO has been forming thin layers in water, while rGO aggregated into large particles [75]. The rGO effect on *E. coli* was significantly lower; thus, uniform dispersion may be one of the crucial factors staying behind antibacterial activity of flake graphene [75]. On the other hand, according to a simulation using *E. coli* and AFM by Romero-Vargas et al. [87], GO rarely penetrates the cell membrane and its interactions are usually repulsive. Moreover, hydrophilic properties cause GO sheets to adhere to the cell without causing any damage [88]. This in turn may result in yet another antibacterial mechanism of GO. In this scenario, GO flakes after adhesion to the cell membrane act as a barrier preventing bacteria from accessing nutrients.

Determination of how much a physical mechanism contributes to the overall antibacterial activity of GO is challenging because of multiple factors (such as different oxidation states of GO and its size) affecting this phenomenon. All of this can be the result of GO structure and morphology. Researchers use graphite to produce graphene from various sources, thus leading to different properties of the final product. Moreover, there is no single method or strict conditions of GO and rGO synthesis followed by their purification. Thus, GO and rGO flakes studied over the world (even though named the same) should be considered as completely varied materials, characterized with different properties.

All of the above also applies to studies about oxidative stress and its participation in antibacterial activity. This mechanism can be divided into two paths: ROS-dependent and ROS-independent. Oxidative stress caused by ROS (reactive oxygen species, e.g., $O_{2}^{•-}$) or by disrupting cellular processes can lead to oxidation of biological compounds and body of cell. This may ultimately result in cell death. A study by Liu et al. [75] showed that small amounts of $O_{2}^{•-}$ by GO or rGO using the XTT method and ROS may play a minor role in antibacterial activity. Nevertheless, GSH (Glutathione) is oxidized in the presence of GO/rGO, while rGO shows an oxidation level similar to the $H_{2}O_{2}$ control trial. On the other hand, Gurunathan et al. [78] measured ROS generation in the extracts from bacterial cells (*P. aeruginosa*) grown in liquid cultures. Results show that GO and rGO produced 3.8-fold and 2.7-fold higher levels of ROS, respectively, in comparison to the control trial. The presence of reduced GSH and NAC (N-Acetylcysteine) reduced elevated ROS generation. Moreover, when antioxidant is added to the cell culture, about 20% more cells of *E. coli* survived incubation with GO [77]. This indicates that other mechanisms beside oxidative stress are important for antibacterial activity of GO. A more systematic and theoretical analysis is required in order to thoroughly establish mechanisms of antimicrobial activity of graphene derivatives. Studies should deeply relate properties of GO and rGO such as size, density of functional groups, chemical composition, and number or layers to the certain mechanism and viability of certain cells.

### 4.4. Antiviral Properties of Graphene Derivatives

Reports about antiviral activity of non-modified GO or rGO are limited. A number of authors have recognized that graphene derivatives and its composites can inhibit cell infections by certain viruses [89,90,91]. Sametband et al. [90] infected Vero cell cultures with the herpes simplex virus type-1 (HSV-1) and tested plaque reduction with 0.5–15 μg/mL GO or modified rGO-SO_3_ added to the virus suspension. A concentration above 5 μg/mL effectively inhibited viral infection while no cytotoxicity towards Vero cells was observed. Another study [89] was conducted using porcine epidemic diarrhea virus (PEDV-RNA virus) and pseudorabies virus (PRV-DNA virus) (Figure 6) and ended up with a similar result with lower levels of GO. A low concentration of GO 6 μg/mL was tested over time and antiviral activity increased. Taken together, these results clearly manifest that concentration of flake graphene should be taken into account when considering its antiviral application. By binding GO with cationic polymer PDDA, Ye et al. [89] found out that the negative charge of GO is responsible for its antiviral behavior. Moreover, the stage at which GO inhibited the virus was investigated. Applying GO and the virus at a different stage showed that GO inhibits virus infection by inactivating the virus before its entry into the cell. Another study [91] tested antiviral activity of GO and GO-Ag against enveloped and non-enveloped viruses. Feline coronavirus (FCoV) served as enveloped (with an outer lipid bilayer) and infectious bursal disease virus (IBDV) without an envelope. Results showed relatively low inhibition of 7.2% against FCoV when GO was used at concentration of 1 mg/mL and an improvement of 1.2% was noted when GO-Ag was used instead. Interestingly, the IBDV virus completely resisted the non-modified GO inhibition. These results indicate that the presence of an outer lipid bilayer plays a significant role in antiviral activity of graphene and more research needs to be done in this field.

## 5. Effects of Modifications by Graphene Derivatives

### 5.1. Efect of Modifications by GO/rGO on the Cellular Response of Mammalian Cells

Azizi et al. [9] emphasized that three-dimensional tissue scaffolds are able to support the function of myocardium exposed to damage, which has great potential in tissue engineering when trying to regenerate this organ. The researchers emphasized that a meticulously designed and modified architecture makes it possible to provide protection to heart muscle cells while promoting repair of damaged tissues. The specially designed and manufactured structure is implanted at the site of the myocardial infarction, so it mechanically supports the muscle, reducing the load on the heart wall, which improves myocardial performance [92]. Myocardial regeneration and improvement in myocardial performance depends largely on the properties of the scaffold, which stimulates the growth and differentiation of stem cells into myocardial cells [93]. Azizi and colleagues created polyurethane scaffolds by the electrospinning method with structural and mechanical properties of myocardium, subjected them to some modifications, and studied the cellular response of stem cells differentiating into myocardial cells [9]. Four types of materials were proposed for analysis, including two modified by the addition of reduced graphene oxide (rGO) nanoparticles: polyurethane fibrous nanomaterials with directed fibers and randomly distributed fibers, and polyurethane fibrous nanomaterials containing reduced graphene oxide (rGO) nanoparticles, with directed fibers and randomly distributed fibers. The researchers observed that the scaffolds modified by the addition of rGO exhibited a slightly reduced average fiber diameter, presumably due to different viscosity of the composite solution in comparison to the pristine polymeric solution. The addition of rGO also increased the Young’s modulus (in the wet state to 168.3 kPa) and strength of the materials. It was observed that fibrous materials with directed fibers had higher elasticity than fibrous materials with randomly directed fibers. The researchers’ analysis showed that the addition of rGO still makes the material properties hydrophobic; so, the researchers increased the hydrophilic properties of the scaffolds using a plasma treatment method. Analysis of the cellular response showed that cell growth is influenced by both the orientation of the fibers and the used modification—better cell adhesion to the substrate was observed on rGO-modified materials. Moreover rGO-modified materials were characterized with uniform cellular distribution. In summary, the authors of [9] have shown that modification of polymer scaffolds with rGO improves cell growth and differentiation. As mentioned earlier, this modification also led to an increase in Young’s modulus and tensile strength so that the resulting scaffolds can be successfully used to treat myocardial degeneration. In addition, cells located on random fibers acquired a rounded morphology, while cells located on aligned nanofibers maintained a polar morphology and accumulated troponin I in a direction almost parallel to the direction of the nanofiber alignment. Moreover, the presence of nanoparticles resulted in a more uniform distribution of cells on the scaffolds’ surface.

Mahmoudifard and colleagues [94] were motivated by research on repair and regeneration of skeletal muscle and other tissues. The researchers worked on the synthesis of composite nanofibers produced from polyaniline and polyacrylonitrile with the addition of camphor sulfuric acid, by electrospinning. An extremely crucial step was the modification of the scaffolds by addition of graphene oxide (GO) and reduced graphene oxide (rGO) to their composition. The morphology, conductivity, and strength of the fibrous scaffolds produced with and without the addition of graphene oxide and reduced graphene oxide were studied. A significant increase in the strength of the scaffolds with the addition of graphene oxide and reduced graphene oxide (Young’s modulus from about 51 MPa to 95 MPa and 63 MPa, respectively) and conductivity (0.24 S cm^−1^ to 0.58 S cm^−1^ and 0.41 S cm^−1^, respectively) compared to scaffolds deprived of flake graphene, was observed. A key step towards biological assessment of resulting scaffolds was the cell culture including stem cells. The researchers observed significantly more cells adhered to scaffolds enhanced with rGO compared to the other scaffolds. However, the cells were less spread compared to those with GO and without the substances, according to Figure 7. Moreover, all markers of skeletal muscle showed a higher expression rate with composite scaffolds compared to scaffolds without graphene. The researchers demonstrated that scaffolds with addition of GO and rGO are biocompatible materials, suitable for applications in tissue engineering and regenerative medicine. An important conclusion that can be drawn from this study is that the addition of rGO provides better conductive properties and increased stiffness. Researchers indicated that such material composition may be conducive to muscle tissue regeneration. It has been reported that the surface specification of the microfiber material, the modification method, the nonwoven modifying agent itself [5], and the conductivity of the scaffolds [94] affect stem cell differentiation.

Liu and his research team [95] produced fibrous scaffolds from a PHA biopolymer by electrospinning. The fibrous scaffolds were functionalized by adding a rGO/Au composite to the solution, which, the authors believed, contributed to the antibacterial properties. The researchers conducted tests to evaluate the effect of the modified scaffolds on Schwann cell migration and proliferation. Tests were conducted on nanofibrous scaffolds with PHA, with different amounts of rGO, and with the addition of an rGO/Au composite. It was noted that scaffolds fabricated from PHA/rGO/Au were the most effective. Electrostimulation was also indicated as a factor triggering higher cell proliferation efficiency. It is known that, as a conducting material, graphene used for modification affects the migration and adhesion of neural cells [96]. A significantly higher number of cells was observed on rGO/Au-modified materials and subjected to electrostimulation than on the other scaffold variants. In conclusion, biological studies showed that conductive nanofibrous scaffolds with a rGO/Au composite clearly promoted the proliferation and migration of Schwann cells. The rGO-modified fibrous scaffolds show potential applications in regeneration and repair of the nervous system [95].

Stone, Lin and Mequanint [97] pointed to the suitable porosity and conductivity as an important property of materials used as scaffolds in tissue engineering. They emphasized that especially in applications involving regeneration of cardiac muscle and nervous system tissues, adequate propagation of electrical signals is required. The researchers set out to develop scaffolds characterized by proper conductivity and porosity. They developed electrospun structures of poly (ester amide) (PEA) and PEA-chitosan, which they modified by adding reduced graphene oxide (rGO). The modifications increased the conductivity of the materials and promoted the differentiation of mesenchymal stem cells into cardiac muscle cells. Gene expression was lower on day 14 compared to day 7, demonstrating that generally a 7-day culture was sufficient. The authors indicated that GATA-4 and Nkx2.5 are both early markers of cardiac differentiation. They highlighted that these markers were upregulated on both rGO-containing scaffolds.

A Taiwanese team of researchers [98] focused their research on cartilage tissue regeneration. The researchers used an electrospinning method to fabricate fibrous scaffolds from poly (L-lactic acid) (PLLA), which were then modified with reduced graphene oxide (rGO) and polydopamine (PDA). The results showed that the electrical conductivity and mechanical properties are improved by the addition of rGO. Moreover, an increase in cell proliferation and secretion of extracellular matrix (ECM) was observed on PLLA/rGO/PDA fibrous scaffolds. The fabricated scaffolds were also exposed to electrostimulation. The viability and distribution of ATDC5 cells, which were cultured on diverse types of scaffolds, were examined. It was observed that the PLLA/rGO scaffold, without the addition of polydopamine, did not promote cell adhesion; so, polydopamine is mainly responsible for the increase in cell adhesion to the substrate in this study. The use of PLLA/rGO/PDA composite scaffold improved proliferation, increased cell adhesion and biocompatibility. Figure 8 depicts fluorescence microscopy images showing the results obtained after 3 days of cell culture on electrically stimulated materials. It was found that the addition of rGO affects the voltage to be applied during the electroporation process, and that low-intensity electrostimulation can affect cell proliferation.

Girao and his team [99] decided to take advantage of the fact that graphene oxide (GO) supports the formation of bioactive neural environments, easily interacts with other biomaterials, and enhances their mechanical properties. Researchers fabricated polycaprolactone-gelatin (PG) fibrous scaffolds by electrospinning for neural tissue engineering applications. They used the fabricated nanofibers to obtain 3D hydrogel architectures. Their results indicate that the addition of graphene oxide (GO), or reduced graphene oxide (rGO), modifies the properties of the fabricated nanofibers and has a significant effect on enteric neural precursor cell (ENPC) proliferation. The nanofibers were placed on 3D constructs and the researchers demonstrated that modification of the fibrous material using rGO with larger size implies the best response from the cells. On the other hand, smaller flakes of reduced graphene oxide cause a decrease in the cytocompatibility of the fabricated nonwovens. The team emphasized that by modifying the chemical composition of the scaffold, it is possible to influence its porosity and mechanical properties. The improved mechanical reinforcement provided by the rGOn nanosheets resulted in an increased tensile modulus in the PG-rGOn nanosheets (60.0 MPa), compared to the values for the PG (48.6 MPa) and PG-GOn (51.0 MPa) electrospun scaffolds. The obtained in vitro results confirm that the presented method of scaffold creation has a remarkable potential in terms of supporting neural networks and is a prelude to further research and that flake graphene can undoubtedly be considered as an attractive material for neural tissue regeneration. The 3D-PG designation stands for hydrogel-material architecture, a material fabricated from polycaprolactone-gelatin, then placed in rGO solution. The 3D-rGO designation denotes a structure in which rGO was also already added to the polymer PG matrix. Figure 9 shows microscopic images of ENPC cells on 3D scaffolds and a graph comparing the area occupied by live and dead cells. In conclusion, the researchers demonstrated that an in vitro neural network culture can be successfully performed on both types of scaffolds. The addition of rGO had a positive effect on responses from ENPC cells—such as adhesion, proliferation, and differentiation.

A group of researchers from South China University of Technology [100] developed a method to produce nanofibrous polycaprolactone (PCL) scaffolds for tissue engineering and biomedical applications. These scaffolds were fabricated by electrospinning and modified by adding different weight concentrations of graphene oxide (GO). The researchers demonstrated that the addition of GO helped to improve the thermal and mechanical properties of the nanostructures, improve the electrical conductivity (from 1.53 µS/cm for 0 wt% GO, to 11.63 µS/cm for 1 wt% GO), and the average diameters of the nanofibers decreased with increasing GO concentration. Furthermore, the researchers analyzed the effect of GO concentration on the cellular response of two selected cell lines: mouse marrow mesenchymal stem cells (mMSCs) and rat pheochromocytoma cells (PC12-L). It was observed that both mMSCs and PC12-L cells successfully adhered to the surface of nanofibrous scaffolds. The mMSCs cells completely covered the surfaces of the nanofibers, and no significant differences in cell growth were observed among material samples with different GO contents. Interestingly, PC12-L cells cultured on materials modified by GO addition showed a better and faster growth. Figure 10 shows the proliferation of mMSCs and PC12-L cells. Song and co-workers stained the fixed cells by using two dyes: cell skeleton by green dye (Cell Navigator™ F-Actin Labeling Kit) and cell nucleus by blue dye (DAPI). Stained cells were observed with confocal microscopy. Analysis of cell adhesion, migration, and proliferation showed that GO addition did not adversely affect proliferation, but facilitated cell adhesion, proliferation, and maturation. Cells showed a morphology typical to fibroblasts (mMSCs) and neurons (PC12-L). The presence of hydrophilic oxygen-containing groups on GO surface and increased fiber roughness contributed to the improved cell growth. Overall, it was found that the morphology of nonwoven fabrics modified by GO addition was more favorable than that of pure PCL, and the fiber diameter decreased with increasing GO concentration, 0.1% and 0.3% GO content improved the mechanical properties of the materials. For the conducted cell cultures, it was noted that the addition of 0.3% and 0.5% was beneficial to the cells, promoted adhesion spreading and maturation. Most importantly, it was found that GO addition significantly increased the differentiation of mMSC and PC12-L cells into osteo- and neurocompatible cells. Expression of β-Tubulin III by PC12-L cells was observed after 6 and 9 days of culture. Positive anti-β-Tubulin III expression was more visible on PCL/GO composite scaffolds with 0.3 wt% GO and 0.5 wt% GO than on other scaffolds. Moreover, staining results of PCL/GO composite scaffolds with 0.3 wt% GO and 0.5 wt% GO were positive for anti-β-Tubulin III and indicating that the addition of GO promoted the differentiation of PC12-L cells into neurons. Researchers proved that electrospun polymer-GO composites have wide applications not only in bone reconstruction and regeneration, but also in nervous system regeneration [100].

In another paper, Wang and co-authors [101] produced fibrous scaffolds for nerve tissue regeneration by electrospinning. The material they used was a composite consisting of Antheraea pernyi silk fibroin (Ap F) and poly (L-lactic acid-co-caprolactone) (PLCL). The electrospun scaffolds were modified with reduced graphene oxide (rGO). The scaffold fabrication process consisted of three steps: (1) electrospinning and crosslinking of Ap F/PLCL nanofibrous scaffolds, (2) immersion of scaffolds in GO solutions with different concentrations (0.5 mg/mL, 1 mg/mL, 1.5 mg/mL, 2 mg/mL), (3) reduction of GO to rGO by immersion in ascorbic acid. The researchers emphasized the appropriateness of these steps, due to the promotion of neuronal cell growth by the graphene derivative content, and the imparting of conductive properties to the materials. The effects of the modifications on morphology, conductivity, hydrophilicity, and mechanical properties were investigated, as well as, more relevant to this review, the effects on migration, proliferation, and myelination of Schwann cells. They investigated the effect of modified scaffolds on P12 cell differentiation using electrostimulation. The growth of Schwann cells and PC12 cells was analyzed on nanofibrous materials modified with and without rGO, some of which were subjected to electrostimulation process. In addition, Sprague–Dawlej (SD) rats were used for in vivo studies. The performed experiments proved that the modification of fibrous scaffolds increased the roughness of fibers. There was also an increase in the average diameter of the fibers—from 518 nm for scaffolds without graphene to 626 nm for scaffolds with 2 mg/mL GO, which indicated the presence of graphene on the surface of nanofibers. Furthermore, the addition of rGO improved the hydrophilic properties of the scaffolds (from 77.9° for scaffolds without graphene to 97.9° for scaffolds with 2 mg/mL GO). The unmodified nanofiber scaffolds showed the lowest mechanical properties compared to the scaffolds with graphene addition (tensile strength about 7 MPa, 9 MPa, 10 MPa, 12 MPa, 14 MPa, for 0, 0.5, 1.0, 1.5 and 2 mg/mL GO, respectively). The rGO coating on the scaffolds provided particularly good conductivity, which ensured the efficiency of the electrostimulation process. It was noted that the morphology of Schwann cells was significantly better on rGO-coated materials than on unmodified materials. The growth and proliferation of these cells was more efficient with electrostimulation than without this process. It was also noted that greater differentiation and better growth of PC12 cells was observed on rGO-modified scaffolds. To evaluate nerve regeneration, three immunohistochemical indices were used, including S-100 and NF-200. Analysis comparing autograft, rGO-modified scaffold, and unmodified scaffold showed that no significant difference was observed between regeneration on autograft and rGO-modified scaffolds. Figure 11 shows the stained S-100 and NF-200 proteins for the regenerated neural tissue in the analyzed groups. It can be seen that regeneration of nerve tissue on rGO-modified scaffolds was comparable to regeneration of nerve tissue on autograft. In contrast, regeneration occurred much worse on unmodified scaffolds. In summary, the work of Wang and coworkers [101] demonstrates that graphene is an extremely promising material for modifying the surface of scaffolds for tissue engineering applications. The two-step rGO modification protocol designed in this study improved mechanical, conductive, and biocompatibility properties. Moreover, the use of electrostimulation promoted the migration and proliferation of Schwann cells and the growth and differentiation of PC12 cells even further. Grafts using rGO-modified scaffolds were proven to contribute to peripheral nerve repair in vivo, at a level similar to autografts.

In a seminal paper written by Ma and coworkers [102], the researchers developed three types of electrospun fibrous scaffolds based on lactic acid polyacid (PLA). They fabricated a scaffold from pure PLA and two types of scaffolds resulted from modification by adding hydroxyapatite (HA) and graphene oxide (GO). Thus, they obtained a PLA scaffold, a PLA/hydroxyapatite (HA) scaffold, and a PLA/HA/graphene oxide (GO) scaffold. The morphology and composition of the nonwovens were studied using a series of analyses to confirm the success of the scaffold fabrication processes. To investigate the cytocompatibility of the fabricated scaffolds, MC3T3-E1 bone cells were cultured on their surface. The researchers evaluated cell adhesion to the surface, proliferation, and differentiation capacity. It was noted that the PLA/HA/GO scaffold exhibited significant roughness. This property promoted cell-surface adhesion and proliferation, which had a positive effect on bone tissue regeneration.

Magaz et al. [103] conducted a study on neural tissue regeneration using electrospun scaffolds fabricated from silk fibroin (SF), which were modified by adding different amounts of graphene oxide (GO) flakes. The properties of the aforementioned nano- and microfibrous scaffolds were fabricated and compared: pure SF, SF/GO, and SF/reduced graphene oxide (rGO) materials were also obtained by in situ reduction. By imparting conductive properties to the scaffolds through the addition of rGO (the presence of rGO increased the conductivity to about 4 × 10^−5^ S cm^−1^ in the dry state and peaked at about 3 × 10^−4^ S cm^−1^ in the hydrated state), electroconductive substrates were obtained that supported neurite growth without the need for external electrostimulation. The presence of GO/rGO did not significantly affect the fiber diameter (about 800 nm) distribution and porosity (70–80%). NG108-15 neuronal cells were able to adhere and grow on each of the fabricated materials but GO-modified materials showed markedly increased proliferation and metabolic activity. The percentage of live cells after 7 days of culture was higher than 95% for all scaffolds.

### 5.2. Effects of Modifications by GO/rGO on Antibacterial Properties

Cojocaru et al. [104] started work on chitosan-based fibrous materials, fabricated by electrospinning. These materials were functionalized with citric acid (CsA), reduced graphene oxide (rGO), and tetraethylenepentamine (TEPA). The researchers employed the biological potential and properties of these materials for medical applications such as wound dressings to aid in the healing process. Three types of materials were produced that differed in rGO/TEPA concentration. The nonwovens were subjected to a series of assays and in vitro cytocompatibility and cytotoxicity tests. After 6 days of culturing NCTC fibroblast cells on rGO/TEPA-modified nonwovens, similar levels of cell proliferation and growth promotion were observed. Figure 12 shows microscopic images showing cells that proliferated and grew in the monolayer adjacent to the substrate. Many of the cells exhibited the desired elongated shape; however, smaller spherical cells that were not adherent to the substrate were also present. The antimicrobial potential of the fibrous materials was evaluated against two strains of bacteria: gram-positive (+) *Staphylococcus aureus* and gram-negative (−) *Pseudomonas aeruginosa*. Biological analysis showed that high viability and elevated levels of cell proliferation were recorded on the fibrous material designated by the researchers as CsA/PGT2, a nonwoven fabric with 0.25 percent rGO/TEPA composite. Moreover, all of the produced fibrous materials exhibited antimicrobial activity.

Pan and co-authors [105] fabricated nanofibrous materials by electrospinning from a solution of poly(ε-caprolactone) (PCL) modified by the addition of silver nanoparticles (Ag) and reduced graphene oxide (rGO). Researchers have reported that biodegradable materials, which include PCL, are undoubtedly widely used, but are often subject to bacterial contamination, which reduces their applicability [106]. The researchers investigated the physicochemical properties of the resulting materials, as well as the antimicrobial properties. Through SEM analysis, it was noted that the fibers in all the materials were randomly distributed, and the addition of rGO/Ag resulted in a decrease in the mean value of fiber diameters. The addition of Ag/rGO improved the strength of the materials. A antibacterial test was conducted using *Staphylococcus aureus* and *E. coli* bacteria. The fabricated fibrous materials were left in contact with the bacterial suspension for a limited time. After the necessary procedures, the number of bacteria present in the samples was recorded to determine the antibacterial efficacy of the fabricated materials. It was shown that after 2 h of contact with Ag/rGO-modified fiber materials, 99.55% and 99.46% of *Staphylococcus aureus* and *E. coli* bacteria were inactivated, respectively. It is theorized that graphene oxide acts as a nano-sharpener to damage the cell membrane of bacteria and causes bacterial death, and the rate of bacterial death depends on the structure of the cell wall [106,107].

Mukheem and his team [108] also worked on the fabrication and characterization of electrospun nanocomposite scaffolds modified with the addition of graphene oxide flakes. In this case, the scientists’ work was a response to the needs of medical professionals concerning the regeneration of the largest organ of the human body, which is the skin. The researchers produced fibrous materials made of polyhydroxyalkanoate (PHA) with the addition of an rGO/Ag composite. The researchers set out to combine the antimicrobial activity of the rGO/Ag nanocomposite with the biocompatible and highly biodegradable PHA and investigated the effects of the fabricated materials on *E. coli* and *Staphylococcus aureus* bacteria. The antimicrobial activity was evaluated at time intervals. It was noted that there was a significant decrease in bacteria on PHA/rGO nonwoven fabric and PHA/rGO/Ag nonwoven fabric compared to nonwoven fabric without rGO or rGO/Ag modification. In conclusion, the researchers successfully fabricated PHA, PHA/rGO, and PHA/rGO/Ag nanocoating. The antibacterial properties of the electrospun scaffolds were tested. It was proved that the PHA/rGO/Ag nanofiber showed significant antimicrobial activity against PHA/rGO and PHA.

Liu and co-workers [109] developed antibacterial nanomaterials for biomedical applications. They fabricated fibrous scaffolds from polylactic acid (PLA) by electrospinning and, to enhance antibacterial and tensile properties of the fibrous scaffolds, the researchers modified its composition by adding of 1 wt% graphene oxide (GO) and 1–7 wt% silver nanoparticle (AgNP). For comparison, the scientists fabricated a matrix made of PLA-1 wt% GO and PLA-AgNP too. The researchers examined the mechanical and antimicrobial properties of the resulting materials. The tests showed that the conducted modifications resulted in significant improvements of mechanical properties, including tensile strength with the improvement of thermal stability at the same time. In addition, the modifications were also shown to promote improved antimicrobial properties in tests against *Esherichai coli* and *Staphylococcus aureus.*

A team headed by Yang [110] has worked on the preparation and characterization of antibacterial nonwoven fabrics made of chitosan (CS), poly(vinyl alcohol) (PVA), and graphene oxide (GO), which have potential applications in regenerative medicine. The researchers emphasized that electrospun fibers, due to their nanometric size, promote drug absorption and water transport [111] which is extremely important in the wound healing process. The antimicrobial activity of nonwovens was tested against *Eschierichia coli* and *Staphylococcus aureus*. Growth inhibition was noted for both *E. coli* and *S. aureus*, so CS/PVA/GO composite nonwovens can be considered as a promising material for skin regeneration applications.

Zhang and his team [20] generated fibrous materials from silk fibroin (SF)/gelatin (GT) composite, which were modified with the addition of graphene oxide (GO) and silver nanoparticles (Ag). The antimicrobial properties of the materials were tested against *E. coli* bacteria. It was observed that the highest inhibition of colony growth occurred on the GO/Ag modified material. A higher number of bacteria was recorded on the GO-only- modified material than on GO/Ag, but the largest colony was observed on the SF/GT control sample. In conclusion, no antimicrobial effect was observed on the SF/GT material, lower *E. coli* counts were observed on the SF/GT/GO material. In contrast, the SF/GT/GO/Ag material showed clear antimicrobial properties and toxicity to *E. coli*. 

Based on all the above-mentioned literature and experimental results, it was possible to claim that graphene and derivatives positively affected the cellular response of several types of mammalian cells and exhibited excellent antibacterial and antifungal properties against microorganisms used in the investigations. In the Table 2. summarized the results of the modifications carried out with graphene on polymer scaffolds, mentioned in the previous section of the manuscript.

The reason graphene was so effective regarding inhibition of bacteria and fungi was probably due to the direct contact with the cell walls of bacteria and fungi. This physical contact can be further translated to the release of the reactive oxygen-species, triggering graphene derivatives reaction with organic functional groups and polysaccharides on the cell walls of bacteria and fungi.

In our opinion, the incorporation of graphene (and derivatives) into the volume of a 3D electrospun fibrous scaffold for regenerative medicine applications is a brilliant idea and the subject of many ongoing studies. Graphene is still a relatively new material, and a lot of research, observation, and clinical trials have to be conducted, but it is likely that its amazing potential will contribute to devising materials that will successfully replace native parts of the living organism.

## 6. Conclusions and Future Remarks

This review presents various concepts for modification of fibrous scaffolds fabricated by electrospinning technique for regenerative medicine and tissue engineering applications. Concepts for obtaining nanofibers from various polymers, which were further modified by GO/rGO addition, are described. It has been reported that studies by many teams of scientists from all over the world clearly indicate that modification of the composition of electrospun scaffolds by the addition of GO/rGO positively affects the cellular response of several types of mammalian cells and imparts antibacterial properties to the fibrous materials. However, some fundamental questions about graphene antimicrobial mechanisms and cytotoxicity of graphene-based nanomaterials, especially during long-term utilization still have to be addressed. The addition of flake graphene to the polymeric matrix of 3D fibrous tissue engineering scaffolds is a promising approach for altering surface properties of resulting materials, triggering desired cellular behavior. To the authors’ best knowledge, fully controllable introduction of flake graphene in regenerative medicine should be preceded by systematic and statistically significant screening of distinct kinds of graphene flakes. This in fact should be treated as an axiom, governing flake graphene fate. Is completely certain that the final effect of modification by GO/rGO depends on many component parts, including oxidation/reduction degree of graphene and its purity, method of graphene production or reducing agent used in the process. Moreover, the size distribution of graphene flakes can vary from a few nanometers to dozens of micrometers. Therefore, rigorous characterization of graphene is needed, before any further implementation is done. This is crucial if any definitive conclusions about graphene-based materials are to be drawn and correlated with antimicrobial properties of resulting materials.

In summary, although according to analyzed papers modifications with GO/rGO greatly improve fibrous bioparameters, future studies need to focus on in vivo studies in order to introduce these materials to modern medicine. Furthermore, many of presented papers are focused on including graphene into polymer, and part of them involved modifications on surface polymer fibrous materials—more research needs to be done with both methods of modifications in the same conditions to compare which way of addition graphene is more effective and gives more desirable properties to scaffolds. In addition, antimicrobial and antiviral mechanisms of GO/rGO still require more research to achieve consensus among scientists. Despite those doubts, it is clear that GO/rGO modifications will play a significant role in the future of modified 3D scaffolds and tissue engineering.

## Figures and Tables

**Figure 1 materials-15-05306-f001:**
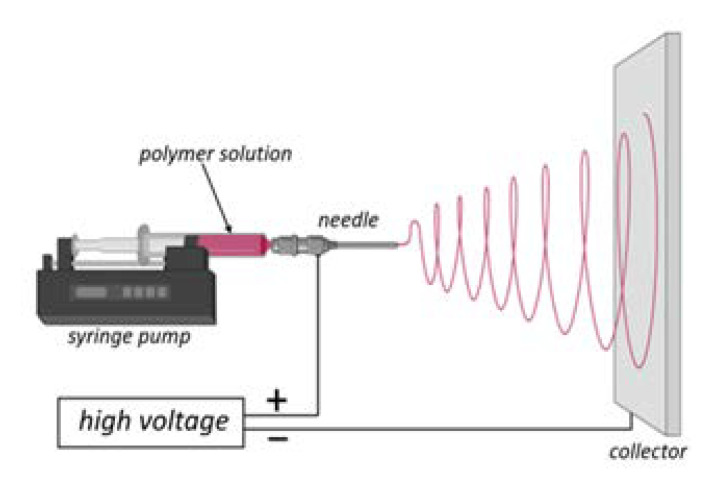
Schematic diagram of the electrospinning setup. Adapted from “FullTemplateName”, by BioRender.com (2022) [28].

**Figure 2 materials-15-05306-f002:**
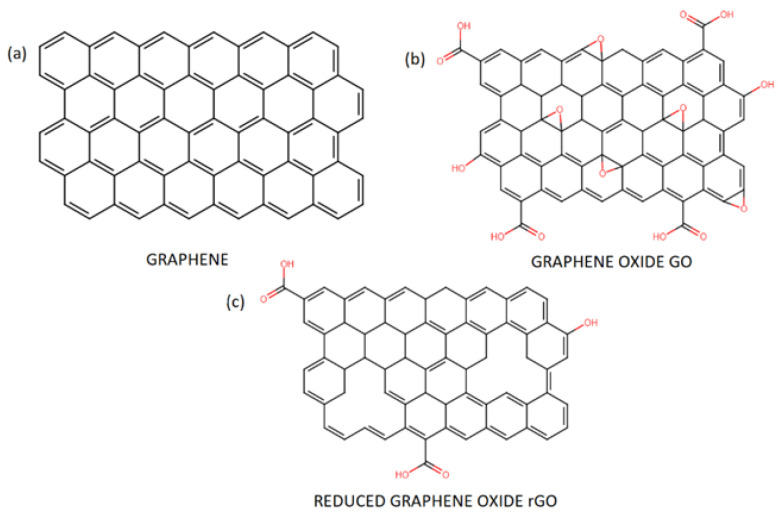
Chemical structure of pristine graphene (**a**) and its derivatives: graphene oxide GO (**b**) and reduced graphene oxide rGO (**c**).

**Figure 3 materials-15-05306-f003:**
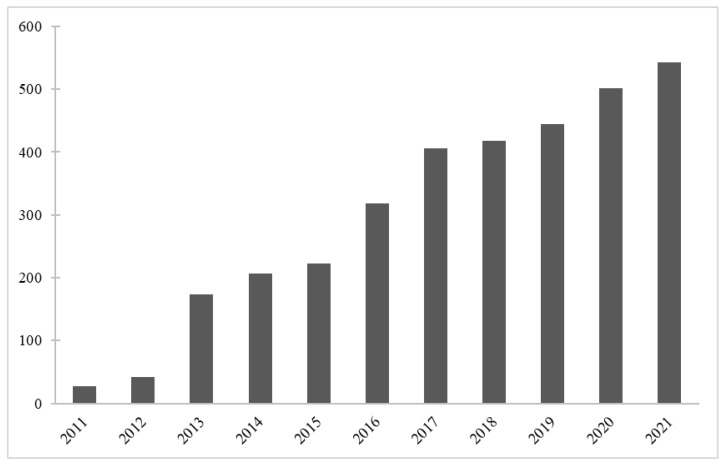
Publications per year on “graphene” in the field of biomaterials. Data retrieved from Web of Science.

**Figure 4 materials-15-05306-f004:**
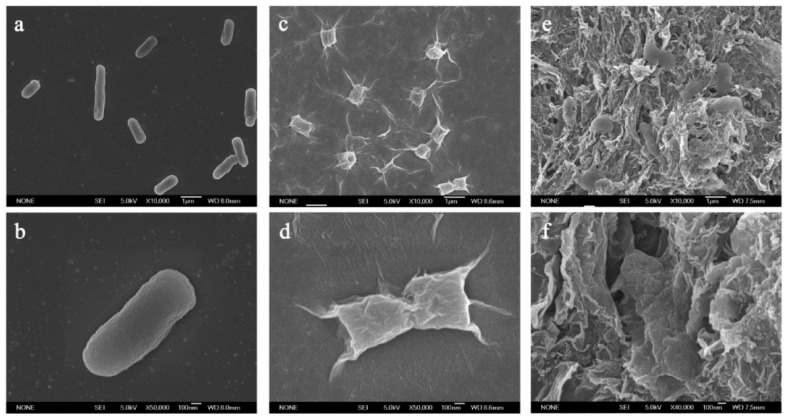
SEM images of E. coli bacteria in (**a**,**b**) saline solution, (**c**,**d**) in GO dispersion (40 µg/mL), (**e**,**f**) in rGO dispersion (40 µg/mL) after 2 h incubation. Reprinted with permission from [75]. Copyright 2011 American Chemical Society.

**Figure 5 materials-15-05306-f005:**
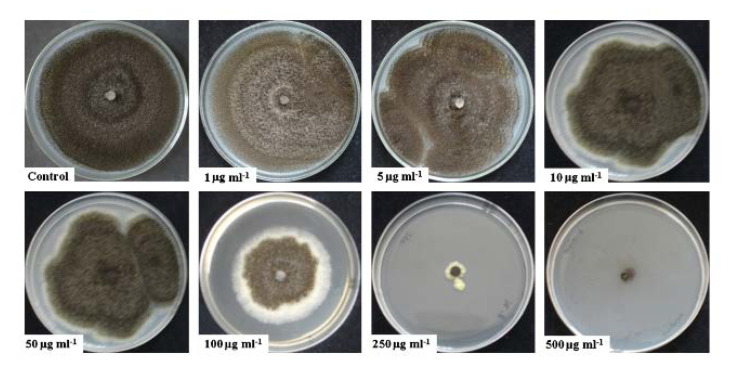
*A. niger* fungi growth in rGO dispersions with different concentrations (0–500 μg/mL) after 7-day incubation. Reprinted from [81] with permission from Elsevier.

**Figure 6 materials-15-05306-f006:**
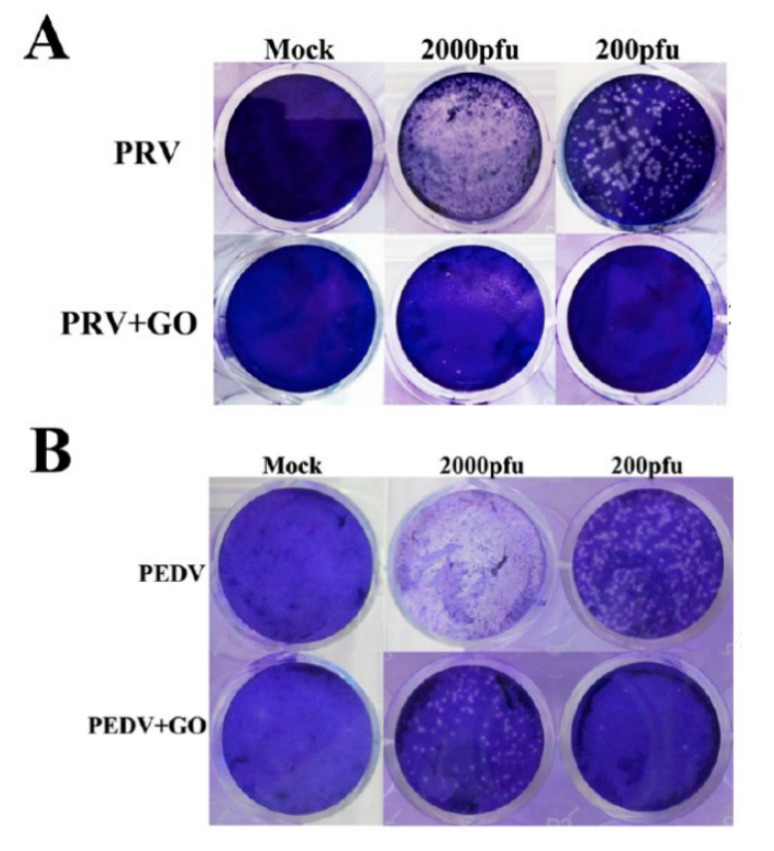
Antiviral activity of GO (6 μg/mL) on PK-15 cells (**A**) and Vero cells (**B**). (**A**) Cells infected with PRV of 200 or 2000 pfu. (**B**) Cells infected with PEDV of 200 or 2000 pfu. Clear spots represent the amount of virus. Mock-infected cells serve as a control. Reprinted with permission from [89]. Copyright 2015 American Chemical Society.

**Figure 7 materials-15-05306-f007:**
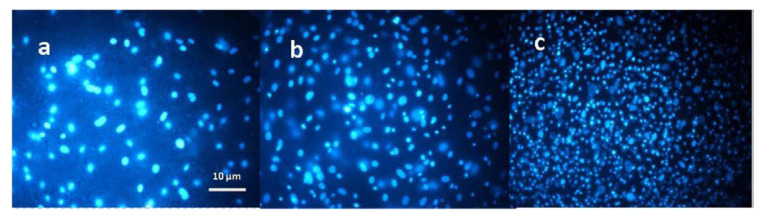
Images of nanofibrous scaffolds with inoculated cells stained with DAPI: (**a**) without modification, (**b**) modified with GO, (**c**) modified with rGO. Reproduced with permission [94] © IOP Publishing.

**Figure 8 materials-15-05306-f008:**
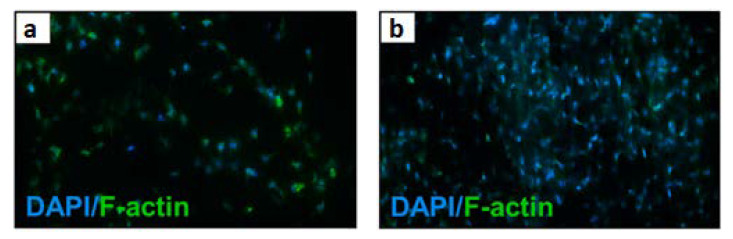
Confocal microscopy images (**a**,**b**) of fibrous scaffolds with ATDC5 cells after three-day culture with piezoelectric stimulation. Reprinted from [98] with permission from Elsevier.

**Figure 9 materials-15-05306-f009:**
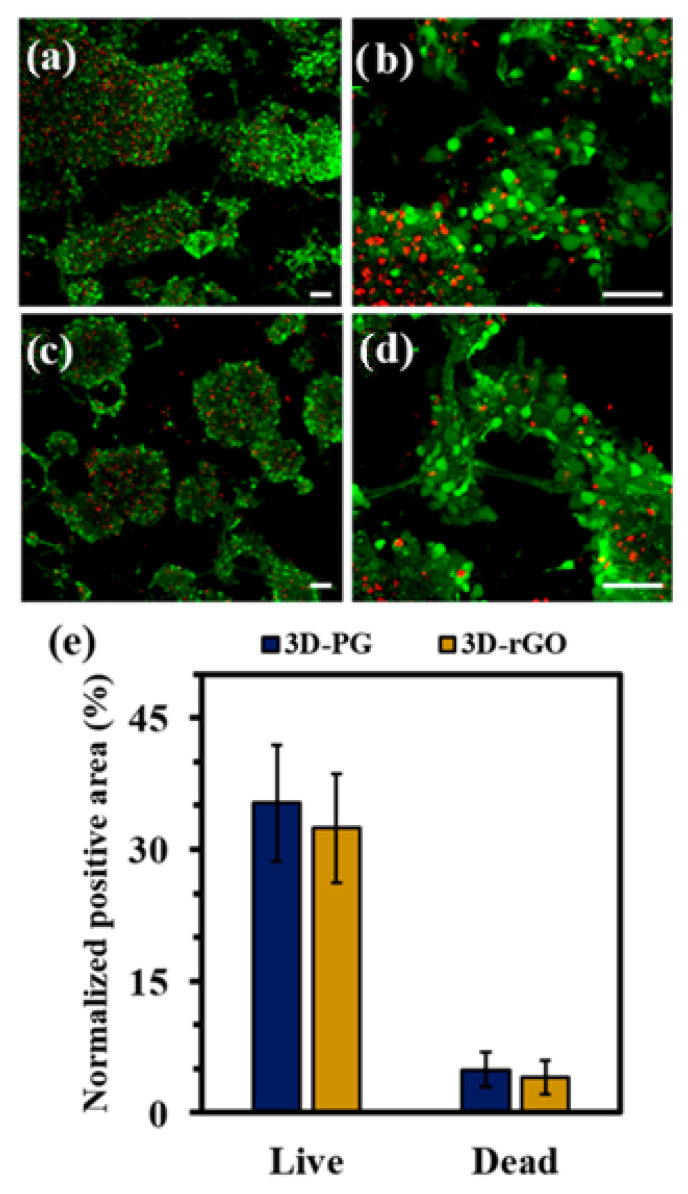
Confocal microscopy images of fibrous scaffolds with ENCP cells after cell culture: (**a**,**b**) fabricated from polycaprolactone-gelatin, then placed in rGO solution, (**c**,**d**) in which rGO was also already added to the polymer PG matrix, (**e**) chart of the area covered by live and dead ENCP cells on the fibrous materials. Reprinted with permission from [99]. Copyright 2022 American Chemical Society.

**Figure 10 materials-15-05306-f010:**
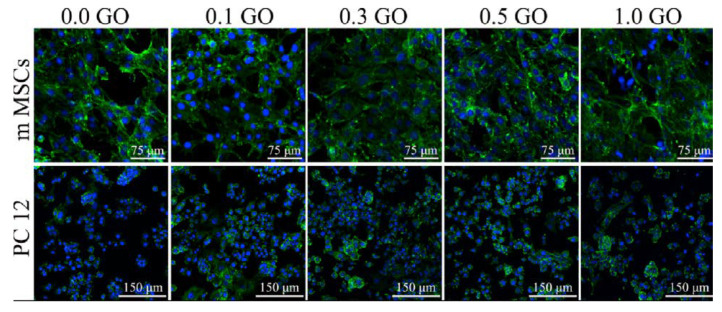
Images of mMSC and PC12-L cell proliferation on PCL scaffolds modified by GO, after 3 days of cell culture with different concentrations of GO (wt%). Reprinted from [100] with permission from Elsevier.

**Figure 11 materials-15-05306-f011:**
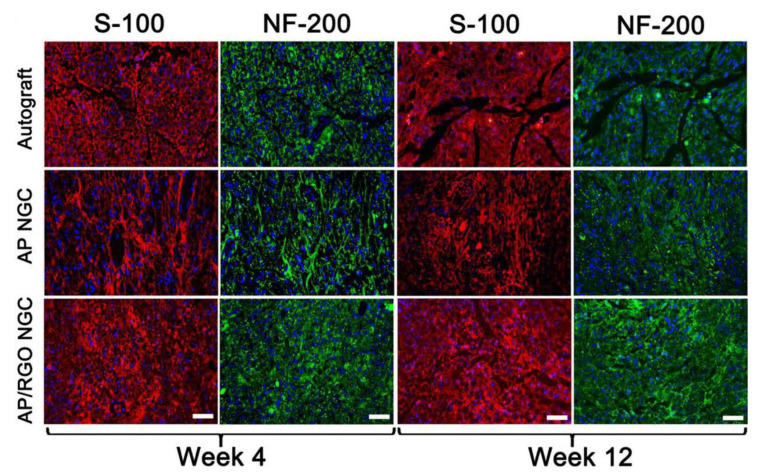
Images of immunofluorescence staining of S-100 and NF-200 in the regenerated nerve tissues (autograft, unmodified scaffold, scaffold modified by rGO). Scale bar equals 50 μm. Reprinted from [101] with permission from Elsevier.

**Figure 12 materials-15-05306-f012:**
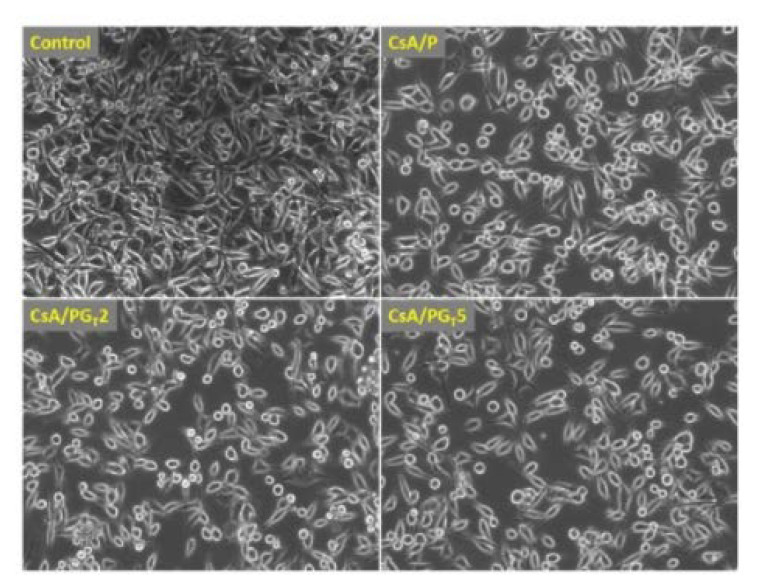
Microscopic images of fibroblasts on fibrous materials, reprinted from [104].

**Table 1 materials-15-05306-t001:** Antimicrobial activity of non-modified graphene derivative.

Material	Studied Organism	Results	Reference
GO	*E. coli*	The loss of viability at: 69.3% after 1 h to 89.7% after 4 h [40 µg/mL]; 91.6% at 1 h [80 µg/mL]	[75]
rGO	*E. coli*	The loss of viability at: 47.4% after 1 h to 74.9% after 4 h [40 µg/mL]; 76.8% after 1 h [80 µg/mL]	[75]
GO/rGO	*E. coli*	The loss of viability at: above 98.5% after 2 h [85 µg/mL]	[76]
GO/rGO paper	*E. coli*	No cell growth on GO paper, some number of E. coli colonies on rGO paper compared to control group	[76]
GO	*E. coli*	The loss of viability is depending on the size of GO sheets; i.e., the smaller the GO sheet, the higher is the loss of viability	[77]
GO	*Mammalian cell line—A549*	The loss of viability at: 30% after 2 h and 50% after 24 h [85 µg/mL], (mild cytotoxicity)	[76]
rGO	*Mammalian cell line—A549*	The loss of viability at: above 75% after 2 h [85 µg/mL] (high cytotoxicity)	[76]
GO	*Pseudomonas aeruginosa*	The loss of viability at: 70% after 2 h to 85% after 4 h [100 µg/mL], above 90% after 2 h [150 µg/mL]	[78]
rGO	*Pseudomonas aeruginosa*	The loss of viability at: 60% after 2 h to 85% after 4 h [100 µg/mL], above 90% after 2 h [150 µg/mL]	[78]
GO	*Ralstonia solanacearum*	The loss of viability at: 60% after 2 h [100 µg/mL], above 90% after 2 h [250 µg/mL]	[79]
rGO	*Ralstonia solanacearum*	The loss of viability at: 5% after 2 h [100 µg/mL], above 15% after 2 h [250 µg/mL]	[79]
GO	*Xanthomonas oryzae pv. Oryzae*	The loss of viability at: 19.4% after 1 h to 66.1% after 4 h [50 µg/mL]; 47.8% after 1 h to 88.6% after 4 h [250 µg/mL]	[80]
rGO	*Xanthomonas oryzae pv. Oryzae*	The loss of viability at: 10.8% after 1 h to 24.8% after 4 h [50 µg/mL]; 12.9% after 1 h to 30.5% after 4 h [250 µg/mL]	[80]
rGO	*A. niger*, *A. oryzae*, *F. oxysporum (fungi)*	Concentrations of rGO above 250 µg/mL almost completely inhibited fungi growth	[81]

**Table 2 materials-15-05306-t002:** GO and rGO functionality related with the regeneration of different tissues.

Field of Regenerative Medicine	Polymer	Cell Type	Modifier Substance	Results	Reference
myocardial regeneration	polyurethane	stem cells	rGO	increased adhesion to substrate, improved proliferation; differentiation of cells into myocardial cells	[9]
skeletal muscle regeneration	polyaniline, polyacrylonitrile	stem cells	GO/rGO	increased adhesion to the substrate, improved cell proliferation, differentiation	[94]
regeneration of the nervous system	polyhydroxyalkanolane (PHA)	Schwann cells	rGO/Au	promoting proliferation and migration	[109]
myocardial regeneration/regeneration of the nervous system	poly(esteramid) (PEA), chitosan	stem cells	rGO	differentiation	[97]
cartilage regeneration	poly(L-lactic acid) (PLLA)	ATDC9 cells	rGO/PDA	improved proliferation, increased cell adhesion and biocompatibility	[98]
regeneration of the nervous system	polycaprolactone-gelatin (PG)	ENPC cells	GO/rGO	promoting proliferation, migration and differentiation	[99]
regeneration of the nervous system/bone regeneration	polycaprolactone	mMSC cells, PC12-L cells	GO/rGO	promoted adhesion, spreading and maturation, significantly increased differentiation	[100]
regeneration of the nervous system	antheraea pernyi silk fibroin (ApF), poly(L-lactic acid-co-caprolactone) (PLCL)	Schwann cells, PC12 cells	GO/rGO	promoted migration and proliferation of Schwann cells and growth and differentiation of PC12 cells	[101]
regeneration of the nervous system	poly(L-lactic acid) (PLLA)	MC3T3-E1 cells	HA/GO	promoting proliferation and adhesion	[102]
regeneration of the nervous system	silk fibroin (SF)	NG108-15 cells	GO/rGO	promote proliferation and metabolic activity	[103]
skin regeneration	chitosan	NCTC fibroblasts	rGO/TEPA	promote cell viability and proliferation	[104]

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
