# Peer review of "Flake Graphene as an Efficient Agent Governing Cellular Fate and Antimicrobial Properties of Fibrous Tissue Engineering Scaffolds—A Review"

_materials, 2022, doi:10.3390/ma15155306_

Round 1
Reviewer 1 Report
Interesting work
Author Response
Dear Reviewer,
We are grateful for all the valuable opinions and acknowledgement of our work. All the changes implemented in the manuscript were highlighted with yellow color.
Respectfully,
Authors

Reviewer 2 Report
In this work, Banasiak and colleagues reviewed the use of GO/rGO on electrospun fibers for biomedical applications. They have revised its use for the design of scaffolds for neural/cartilage applications, without and with electrical stimulation, that are both electroconductive and have antibiotic/antiviral activity.
Overall, I believe the manuscript is well written, the information is smartly organized and the authors provide a (yet) brief critical analysis on the available literature. Nevertheless, some key issues can be addressed to make this manuscript more appealing to a general audience. As such, I advise this manuscript to be returned to the authors with Major Revisions needed. Please consider the following:
1. I believe in section 2.2. you could also expand on the sustainability of graphene oxide production from organic waste as a key advantage. Consider the following manuscripts as an example of such:
a) https://doi.org/10.1021/acssuschemeng.8b03997
b) https://doi.org/10.1016/j.matchemphys.2020.122692
c) https://doi.org/10.3390/nano7070182
2. In section 2.2, can you further elaborate on the electrical conduction mechanism of rGO?
3. In phrase 32-33 of page 2, the authors claim that PBS might promote GO aggregation and suppress anti-bacterial activity. All biological fluids contain ions similar to those present in PBS. Please elaborate on the consequences of this for biomaterial design and/or future use of GO/rGO in therapeutics.
4. In phrase 73 of page 4, consider switching “food” for “nutrients”.
5. Overall, I believe section 2.4. has well-balanced text, but it lacks quantitative information that could support some of the claims revised. Please consider adding quantitative information regarding (at least) electroconductivity of the scaffolds, mechanical properties (p.e. Youngs modulus), adhesion rate, marker expression and so forth.
6. In figure 10, can the authors explain what stainings were used?
7. Can the authors provide a better quality replacement for Figure 11?
8. Can the authors check the beginning of phrase 354 in page 13: “The researchers was advised to”.
9. Can the authors clarify the information contained in phase 385-388 on page 14? I thought this hypothesis was debunked previously. I believe bridging this information with the one presented in the previous sections would be valuable.
10. Please check this section of the text, contained in phrases 406-407 on page 15: “oxide-silver GO/Ag nanoparticles”. What are the authors trying to convey here?
Author Response
Dear Reviewer,
We are grateful for all the valuable opinions and acknowledgement of our work. All the changes implemented in the manuscript were highlighted with yellow color.
Respectfully,
Authors
Reviewer #2: 1. I believe in section 2.2. you could also expand on the sustainability of graphene oxide production from organic waste as a key advantage. Consider the following manuscripts as an example of such:
- a) https://doi.org/10.1021/acssuschemeng.8b03997
- b) https://doi.org/10.1016/j.matchemphys.2020.122692
- c) https://doi.org/10.3390/nano7070182
Ad. 1. The section was edited accordingly to the Reviewer’s suggestions:
“Solution to profitability issue and advantage for graphene utilization can be synthesis of GO/rGO from waste materials, such as waste toner powder [Ref. a)], waste newspaper [Ref. b)] and oil palm waste (kernel shells, leaves and empty fruit bunches) [Ref. c)].”
Reviewer #2: 2. In section 2.2, can you further elaborate on the electrical conduction mechanism of rGO?
Ad. 2. The section was edited accordingly to the Reviewer’s suggestions and short note about electrical conduction mechanism was added.
“…due to zero-overlap semimetal with electron and holes as charge carriers [DOI: 10.1016/B978-0-12-814548-7.00016-7].”
Reviewer #2: 3. In phrase 32-33 of page 2, the authors claim that PBS might promote GO aggregation and suppress anti-bacterial activity. All biological fluids contain ions similar to those present in PBS. Please elaborate on the consequences of this for biomaterial design and/or future use of GO/rGO in therapeutics.
Ad. 3. At this point we are not able to predict consequences of implementation of such materials into human body. The issue addressed by the reviewer is in fact related with the ongoing scientific debate about graphene-based materials degradability, stability. In fact, up to date a little is known about the properties of graphene and its derivatives subjected to human body conditions, especially in a long -term relation. We are aware that this issue is important but as stated earlier at this point we don’t want to speculate on this manner.
Reviewer #2: 4. In phrase 73 of page 4, consider switching “food” for “nutrients”.
Ad. 4. The phrase was edited accordingly to the Reviewer’s suggestions.
Reviewer #2: 5. Overall, I believe section 2.4. has well-balanced text, but it lacks quantitative information that could support some of the claims revised. Please consider adding quantitative information regarding (at least) electroconductivity of the scaffolds, mechanical properties (p.e. Youngs modulus), adhesion rate, marker expression and so forth.
Ad. 5. The quantitative information was added where possible accordingly to the Reviewer’s suggestions.
Reviewer #2: 6. In figure 10, can the authors explain what stainings were used?
Ad. 6. The explanation was added accordingly to the Reviewer’s suggestions: “[…] Song and the researchers stained the fixed cells with the two dyes: cell skeleton by green dye (Cell Navigator™ F-Actin Labeling Kit) and cell nucleus by blue dye (DAPI). Stained cells were observed with a confocal microscopy.”
Reviewer #2: 7. Can the authors provide a better quality replacement for Figure 11?
Ad. 7. A photo was added in the best possible quality accordingly to the Reviewer’s suggestions.
Reviewer #2: 8. Can the authors check the beginning of phrase 354 in page 13: “The researchers was advised to”.
Ad. 8. The sentence was changed accordingly to the Reviewer's suggestions.
Reviewer #2: 9. Can the authors clarify the information contained in phase 385-388 on page 14? I thought this hypothesis was debunked previously. I believe bridging this information with the one presented in the previous sections would be valuable.
Ad. 9. Due to technical issue we are not able to find mentioned phrase. If possible, please cite appropriate paragraph from the manuscript and we will try to revise the manuscript once more.
Reviewer #2: 10. Please check this section of the text, contained in phrases 406-407 on page 15: “oxide-silver GO/Ag nanoparticles”. What are the authors trying to convey here?
Ad. 10. Thank you for pointing out the mistake - the description was changed accordingly to the Reviewer's suggestions: "[...] They fabricated fibrous scaffolds from polylactic acid (PLA) by electrospinning and to enhance antibacterial properties and tensile of the fibrous scaffolds researchers modified its matrix composition by adding 1% wt. graphene oxide (GO) and 1–7% wt. silver nanoparticle (AgNP). For comparison, scientists fabricated matrix made of PLA-1% wt. GO and PLA-AgNP too.".

Reviewer 3 Report
It was a review paper about the application of graphene-based materials in the structure of fibrous scaffolds to improve the antibacterial activity of the scaffold and guide the cellular fate for the aim of tissue engineering applications. Here are some comments on this study that should be considered before publication:
1- The quality of abstract need to be improved.
2- “Graphene, a novel 2D carbonaceous material, presents very interesting and unique properties which might be appealing for previously mentioned fibrous scaffolds, leading to lower chance for infection and higher cellular adhesion. We summarized state of research on antimicrobial and antiviral activities of GO/rGO, highlighting any inaccuracies and underlining questions for further studies. The subject of this manuscript gives the Reader an opportunity to follow not only recently developed composite materials, but also their utilization as an efficient tool of personalized modern medicine.” Please add references related to this paragraph.
3- There are several grammatical mistakes in the text that should be corrected.
4- Page 6 lines 41-52 should be deleted because the title of the subheading is about bacterial and fungal cells.
5- Please describe more about the antifungal activity of graphene (and its derivatives) in section 2.3.1.
6- The title of the manuscript should also be change. You mention just antibacterial activity in title while you describe about the antifungal and antiviral properties of graphene in the main text as well.
7- Figure 8 need to be redrawn. Moreover, please rewrite figure caption of figure 8.
8- Please summarize the sample mention in sections 2.4 and 2.5, and add your decision about that, instead of mentioning all results from the mentioned paper.
9- What is the difference between sections a to d in figure 9? You should mention it in figure caption.
10- In samples mention in lines 352-369 could rGO samples affect antibacterial results?
11- In sample mention in lines 371-403, does the antibacterial effect in this sample associated to the Ag NPs or graphene? Please check them again. The same for samples in lines 404-412 and 422-430.
12- Please add a section about the challenges of utilizing graphene based materials.
Author Response
Dear Reviewer,
We are grateful for all the valuable opinions and acknowledgement of our work. All the changes implemented in the manuscript were highlighted with yellow color.
Respectfully,
Authors
Reviewer #3: 1. The quality of abstract needs to be improved.
Ad. 1. Authors checked abstract again and did not introduce any changes to it. We believe that the current version of the abstract is well related with the text of manuscript and does not need to be improved.
Reviewer #3: 2. “Graphene, a novel 2D carbonaceous material, presents very interesting and unique properties which might be appealing for previously mentioned fibrous scaffolds, leading to lower chance for infection and higher cellular adhesion. We summarized state of research on antimicrobial and antiviral activities of GO/rGO, highlighting any inaccuracies and underlining questions for further studies. The subject of this manuscript gives the Reader an opportunity to follow not only recently developed composite materials, but also their utilization as an efficient tool of personalized modern medicine.” Please add references related to this paragraph.
Ad. 2. The references were added accordingly to the Reviewer’s suggestions.
Reviewer #3: 3. There are several grammatical mistakes in the text that should be corrected.
Ad. 3. Manuscript was checked and some errors was removed.
Reviewer #3: 4. Page 6 lines 41-52 should be deleted because the title of the subheading is about bacterial and fungal cells.
Ad. 4. The lines were deleted from this section accordingly to the Reviewer’s suggestions. New section related exclusively with cytotoxicity of graphene derivatives (2.3.2.) was added to main text of manuscript.
“2.3.2. Cytotoxicity of graphene derivatives
Cytotoxicity of graphene materials was also studied and GO induced lower loss of mammalian cell viability then rGO [70]. It is worth to note that lower cell viability occurred due to decreased proliferation rates, not to apoptosis or death of cells like in the case of bacteria. Toxicity of graphene materials in in vitro and in vivo studies was broadly reviewed in Lalwani et al. paper [79]. It was reported that it is highly dependent on parameters, such as time, cell type, size, purity, amount of oxygen functional groups and morphology of graphene. Even though majority of studies show that GO and rGO flakes are cytotoxic towards bacteria and fungi the overriding conclusion might be that specific applications will need separate studies and previous research might be used only as a guide. In fact, it is easy to notice that opposite opinions on graphene cytotoxicity have emerged. Further studies are definitely needed in order to set new frames of knowledge regarding the aforementioned subject.
Scientist [80] highlighted that despite of many advantages and exciting results of using graphene in biomedical engineering and tissue engineering, there may still be a long way ahead before actual application of this material in a clinical practice. The further biological applications of graphene have been often challenged by concerns regarding its potential cytotoxicity. However, authors [80] emphasized that graphene, with “nano-small” sizes, subjected to adequate purification methods, can be implemented as a biocompatible surface coatings and is characterized with brilliant stability in physiological environments, seems to be much less damaging regarding both in vitro and in vivo studies. Moreover, the future focus should be placed on research leading to answers on how to abolish toxicity and affect the degradation of graphene in biological systems of living organisms. Before graphene and graphene-enriched materials can be used in clinical practice, complex studies are needed to resolve such safety issues.
Figure 4. SEM images of E. coli bacteria in (a,b) saline solution, (c,d) in GO dispersion [40 µg/ml], (e,f) in rGO dispersion [40 µg/ml] after 2h incubation. Reprinted with permission from [69] Copyright 2011 American Chemical Society.”
Reviewer #3: 5. Please describe more about the antifungal activity of graphene (and its derivatives) in section 2.3.1.
Ad. 5. More information about the antifungal activity of graphene (and its derivatives) was added accordingly to the Reviewer’s suggestions.
“Antifungal effects of rGO was studied using A. niger, A. oryzae, F. oxysporu. After 7-day incubation growth of all fungi was completely stopped above 250 μg/ml concentration of rGO [75] (Figure 5). In their study, Al-Thani and co-authors [77] prepared GO by a modified Hummers method and characterized it by different techniques. The XRD analysis of graphite powder shown a highly ordered structure, that corresponds to an interlayer spacing of about 0.335 nm. To study the antifungal activities of GO, the material was tested against eukaryotic fungus - Candida albicans. This eukaryotic fungus is characterized by cell structure and metabolism hard to suppress by any antimicrobial agent. In this study, the loss of viability was increased with incubation time of analysis microorganisms. Results revealed that GO has antifungal activity against microorganisms used in this investigation. To summarize, the developed GO exhibited excellent antifungal properties. Sawangphurk and co-workers [78] emphasized that antibacterial activities of graphene and its derivatives had been sufficiently investigated but their antifungal properties were far less studied. In their work, they studied the antifungal activity of rGO against three fungi: Aspergillus oryzae, Aspergillus niger, Fusarium oxysporum. Authors highlighted that graphene (and its derivatives) is of interest due to its high surface area (about 2630 m2 g−1), high electrical conductivity (about 2000 S cm−1), high thermal conductivity (about 4840–5300 W m−1 K−1) and high Young modulus (about 10 TPa) what leads to various potential applications. Reduced graphene oxide (rGO) was produced with a modified Hummers method. The graphene was dispersed in agar and poured into sterilized Petri dishes. Agar discs were covered of fungal, next discs were placed aseptically in the center of agar plates containing rGO nanosheets (0-500 μg ml -1). Experiments were performed for 7 days in five replicates. The average diameters of fungal colonies were determined. The growth inhibition of A. niger , F. oxysporum and A. oryzae was proportional to the concentrations of rGO flakes. The reason of the rGO flakes were effective to inhibit fungi was probably due to the direct contact with the cell walls of analyzed fungi. Scientists reported [78] the rGO inhibited the mycelial growth of the fungi and they hypothesized that it was due to its sharp edges. Upon contact, the reactive oxygen-containing functional groups in several small rGO nanospheres were able to chemically react with the organic functional groups of chitin and other polysaccharides on the fungal cell walls. The half maximal inhibitory concentration (IC50) is a measure of the effectiveness of the rGO in inhibiting the fungi. Authors reported that IC50 values of the rGO against F. oxysporum, A. niger, and A. oryzae were 50, 100, and 100 μg ml−1, respectively. According to the results, the fungitoxicity of rGO against analyzed pathogenic fungi might support the possibility of using rGO as an antifungal nanomaterial.”
Reviewer #3: 6. The title of the manuscript should also be change. You mention just antibacterial activity in title while you describe about the antifungal and antiviral properties of graphene in the main text as well.
Ad. 6. The title was edited accordingly to the Reviewer’s suggestions.
“Flake graphene as an efficient agent governing cellular fate and antimicrobial properties of fibrous tissue engineering scaffolds. A review.”
Reviewer #3: 7. Figure 8 need to be redrawn. Moreover, please rewrite figure caption of figure 8.
Ad. 7. The figure and figure caption was edited accordingly to the Reviewer’s suggestions:
“Figure 8. Confocal microscopy images of fibrous scaffolds with ATDC5 cells after three-day culture with piezoelectric stimulation. Reprinted from [72] with permission from Elsevier.”
Reviewer #3: 8. Please summarize the sample mention in sections 2.4 and 2.5, and add your decision about that, instead of mentioning all results from the mentioned paper.
Ad. 8. The summarize and our opinion about graphene (and derivatives) and graphene modifications of fibrous scaffolds was added accordingly to the Reviewer’s suggestions: “Based on all above-mentioned literature and experimental results it was possible to claim that graphene and derivatives positively affected the cellular response of various types of mammalian cells and exhibited excellent antibacterial and antifungal properties against microorganisms used in investigation. The reason why graphene was so effective regarding inhibition of bacteria and fungi was probably due to the direct contact with the cell walls of bacteria and fungi. This physical contact can be further translated to release of the reactive oxygen-species, triggering graphene derivatives reaction with organic functional groups and polysaccharides on the cell walls of bacteria and fungi.
In our opinion, the incorporation of graphene (and derivatives) into the volume of a 3D electrospun fibrous scaffold for regenerative medicine applications is a brilliant idea and the subject of many ongoing studies. Graphene is still a relatively new material and a lot of research, observation and clinical trials have to be conducted, but it is likely that its amazing potential will contribute to devising materials that will successfully replace native parts of the living organism.”
Reviewer #3: 9. What is the difference between sections a to d in figure 9? You should mention it in figure caption.
Ad. 9. The figure caption was edited accordingly to the Reviewer’s suggestions:
“Figure 9. Confocal microscopy images of fibrous scaffolds with ENCP cells after cell culture: (a-b) fabricated from polycaprolactone-gelatin, then placed in rGO solution, (c-d) in which rGO was also already added to the polymer PG matrix, (e) chart of the area covered by live and dead ENCP cells on the fibrous materials. Reprinted with permission from [73] Copy-right 2022 American Chemical Society”
Reviewer #3: 10. In samples mention in lines 352-369 could rGO samples affect antibacterial results?
Ad. 10. rGO mentioned in these lines was implemented by the authors of the cited articles as an antibacterial agent. Scientific literature have proven that rGO flakes sauté are characterised by antimicrobial properties what is shown in Table 1 of the manuscript.
Table 1. Antimicrobial activity of non-modified graphene derivative.
Material |
Studied organism |
Results |
Reference |
GO |
E. coli |
The loss of viability at: 69.3% after 1h to 89.7% after 4 h [40 µg/ml]; 91.6% at 1h [80 µg/ml] |
[69] |
rGO |
E. coli |
The loss of viability at: 47.4% after 1h to 74.9% after 4 h [40 µg/ml]; 76.8% after 1h [80 µg/ml] |
[69] |
GO/rGO |
E. coli |
The loss of viability at: above 98.5% after 2h [85 µg/ml]. |
[70] |
GO/rGO paper |
E. coli |
No cell growth on GO paper, some number of E. coli colonies on rGO paper compered to control group. |
[70] |
GO |
E. coli |
The loss of viability is depending on the size of GO sheets; i.e. the smaller is GO sheet, the higher is loss of viability |
[71] |
GO |
Mammalian cell line - A549 |
The loss of viability at: 30% after 2 h and 50% after 24 h [85 µg/ml], (mild cytotoxicity). |
[70] |
rGO |
Mammalian cell line - A549 |
The loss of viability at: above 75% after 2 h [85 µg/ml] (high cytotoxicity). |
[70] |
GO |
Pseudomonas aeruginosa |
The loss of viability at: 70% after 2 h to 85% after 4 h [100 µg/ml], above 90% after 2 h [150 µg/ml] |
[72] |
rGO |
Pseudomonas aeruginosa |
The loss of viability at: 60% after 2 h to 85% after 4 h [100 µg/ml], above 90% after 2 h [150 µg/ml] |
[72] |
GO |
Ralstonia solanacearum |
The loss of viability at: 60% after 2 h [100 µg/ml], above 90% after 2 h [250 µg/ml] |
[73] |
rGO |
Ralstonia solanacearum |
The loss of viability at: 5% after 2 h [100 µg/ml], above 15% after 2 h [250 µg/ml] |
[73] |
GO |
Xanthomonas oryzae pv. Oryzae |
The loss of viability at: 19.4% after 1 h to 66.1% after 4 h [50 µg/ml]; 47.8% after 1 h to 88.6% after 4 h [250 µg/ml] |
[74] |
rGO |
Xanthomonas oryzae pv. Oryzae |
The loss of viability at: 10.8% after 1 h to 24.8% after 4 h [50 µg/ml]; 12.9% after 1h to 30.5% after 4 h [250 µg/ml] |
[74] |
rGO |
A. niger, A. oryzae, F. oxysporum (fungi) |
Concentrations of rGO above 250 µg/ml almost completely inhibited fungi growth. |
[75] |
Reviewer #3: 11. In sample mention in lines 371-403, does the antibacterial effect in this sample associated to the Ag NPs or graphene? Please check them again. The same for samples in lines 404-412 and 422-430.
Ad. 11. This is very good question. In fact antibacterial properties of composite of graphene and Ag NPs can driven by each of the material itself or it can be synergistic effect of both of them. In works cited by us, authors focused on fabrication and antibacterial properties examination of graphene/Ag NPs composites. Such composites were fabricated in order to ensure synergistic effect of both materials.
Reviewer #3: 12 Please add a section about the challenges of utilizing graphene based materials.
Ad. 12. Authors did not add separate section but added a few sentences in chapter 3:
“Moreover, size distribution of graphene flakes can vary from a few nanometres to dozens of micrometres. Therefore rigorous characterization of to be utilized graphene material is needed, before any further implementation into other materials is done. This is crucial if any definitive conclusions about graphene based materials are to be drawn and correlated to starting material properties.”

Reviewer 4 Report
Comments: Overall, the flow of the manuscript is very good and the article is suitable to the overall goal of Material /MDPI and this would add value if revised properly.
1- REFs should have been revised according to format style of Material/MDPI journal (Author 1, A.B.; Author 2, C.D. Title of the article. Abbreviated Journal Name Year, Volume, page range).
2- In the text, reference numbers should be placed in one square brackets [ ].
3- The manuscript would greatly benefit from further editing and extensive English check.
4- The work is not novel because it was studied previously. Authors should to describe in deep the novelty of the work compared to other published manuscripts in the same field.
5- The authors should clarify If some formulations have reached clinical application and discuss the clinical results achieved.
6- Cytotoxicity of such these nanomaterials was not exactly highlighted. The topic is so interesting for readier, for this reason, it is suggested if cytotoxicity of these nanomaterials were collected in subsection.
Author Response
Dear Reviewer,
We are grateful for all the valuable opinions and acknowledgement of our work. All the changes implemented in the manuscript were highlighted with yellow color.
Respectfully,
Authors
Reviewer #4: 1. REFs should have been revised according to format style of Material/MDPI journal (Author 1, A.B.; Author 2, C.D. Title of the article. Abbreviated Journal Name Year, Volume, page range).
Ad. 1. References were revised according to format style of Material/MDPI journal accordingly to the Reviewer's suggestions.
Reviewer #4: 2. In the text, reference numbers should be placed in one square brackets [ ].
Ad. 2. Reference numbers were placed in one square brackets according to format style of Material/MDPI journal accordingly to the Reviewer's suggestions.
Reviewer #4: 3. The manuscript would greatly benefit from further editing and extensive English check.
Ad. 3. Authors spellchecked manuscript for any typos and grammatical errors.
Reviewer #4: 4. The work is not novel because it was studied previously. Authors should to describe in deep the novelty of the work compared to other published manuscripts in the same field.
Ad. 4. The Introduction was edited accordingly to the Reviewer’s suggestions in order to describe novelty of this work.
“Several brilliant reviews concerning tissue engineering scaffolds have been published [15,16], but there is a lack of comprehensive summary of graphene influence on antimi-crobial properties of fibrous tissue engineering scaffolds.”
“Graphene, a comparatively novel 2D carbonaceous material, presents very interesting and unique properties which might be appealing for previously mentioned fibrous scaffolds, leading to lower chance for infection and higher cellular adhesion [17]. We summarized state of research on antimicrobial and antiviral activities of GO/rGO, highlighting any in-accuracies and underlining questions for further studies. The subject of this manuscript gives the Reader an opportunity to follow not only recently developed composite materials and their properties, but also utilization of such materials as an efficient tool of personal-ized modern medicine [17–19]”
“Lastly, we described challenges of graphene-based materials utilization and emphasized importance of developing new scientific methods in order to effectively characterize such 2D material and evaluate its influence on selected properties of fibrous scaffolds in tissue engineering, especially regarding their surface.”
Reviewer #4: 5. The authors should clarify If some formulations have reached clinical application and discuss the clinical results achieved.
Ad. 5. Authors were unable to find any clinical results concerning graphene utilized in fibrous tissue engineering scaffolds. We believe that such situation is related with high novelty of described subject.
Reviewer #4: 6. Cytotoxicity of such these nanomaterials was not exactly highlighted. The topic is so interesting for readier, for this reason, it is suggested if cytotoxicity of these nanomaterials were collected in subsection.
Ad. 6. The manuscript was edited accordingly to the Reviewer’s suggestions – new section was added.

Round 2
Reviewer 2 Report
The authors have addressed all issues. I believe the manuscript can be published at its present form.
Reviewer 3 Report
Thanks for addressing the comments.
Reviewer 4 Report
Manuscript was revised point by point according to reviewer comments and It is more acceptable NOW.